# Outcomes of Capecitabine and Temozolomide (CAPTEM) in Advanced Neuroendocrine Neoplasms (NENs)

**DOI:** 10.3390/cancers12010206

**Published:** 2020-01-14

**Authors:** Katharine Thomas, Brianne A. Voros, Meghan Meadows-Taylor, Matthew P. Smeltzer, Ryan Griffin, J. Philip Boudreaux, Ramcharan Thiagarajan, Eugene A. Woltering, Robert A. Ramirez

**Affiliations:** 1Division of Hematology/Oncology, Louisiana State University Health Science Center, New Orleans, LA 70112, USA; ktho24@lsuhsc.edu; 2Department of Surgery, Louisiana State University Health Sciences Center, New Orleans, LA 70112, USA; briannevoros@gmail.com (B.A.V.); jboudr4@lsuhsc.edu (J.P.B.); rthiag@lsuhsc.edu (R.T.); ewolte@lsuhsc.edu (E.A.W.); 3Ochsner Medical Center-Kenner, Neuroendocrine Tumor Clinic, Kenner, LA 70065, USA; 4Department of Epidemiology and Biostatistics, University of Memphis School of Public Health, Memphis, TN 20910, USA; mbmadows@memphis.edu (M.M.-T.); msmltzer@memphis.edu (M.P.S.); 5Ochsner Medical Center, Department of Internal Medicine, Division of Hematology/Oncology, New Orleans, LA 70122, USA; rgriffin@ochsner.org

**Keywords:** Capecitabine, chemotherapy, neuroendocrine neoplasms, temozolomide

## Abstract

Capecitabine and temozolomide (CAPTEM) have shown promising results in the treatment of neuroendocrine neoplasms (NEN). The aim of this study was to evaluate the outcome and role for CAPTEM in malignant neuroendocrine neoplasms. Data were obtained from NEN patients who received at least one cycle of CAPTEM between November 2010 and June 2018. The average number of cycles was 9.5. For analysis, 116 patients were included, of which 105 patients (91%) underwent prior treatment. Median progression free survival (PFS) and overall survival (OS) were 13 and 38 months, respectively. Overall response rate (ORR) was 21%. Disease control rate (DCR) was 73% in all patients. PFS, median OS, ORR, and DCR for pancreatic NENs (pNEN) vs. non-pNEN was 29 vs. 11 months, 35 vs. 38 months, 38% vs. 9%, and 77% vs. 71%, respectively. Patients with pNEN had a 50% lower hazard of disease progression compared to those with non-pNEN (adjusted Hazard Ratio: 0.498, *p* = 0.0100). A significant difference in PFS was found between Ki-67 < 3%, Ki-67 3–20%, Ki-67 > 20–54%, and Ki-67 ≥ 55% (29 vs. 12 vs. 7 vs. 5 months; *p* = 0.0287). Adverse events occurred in 74 patients (64%). Our results indicate that CAPTEM is associated with encouraging PFS, OS, and ORR data in patients with NENs.

## 1. Introduction

Neuroendocrine neoplasms (NENs) are solid tumors with a variable disease course, based on the histology and location of tumor. Currently, NENs are classified by morphological features and proliferation rate [1,2,3]. The World Health Organization (WHO) classification for NENs defines subgroups by both mitotic count and Ki-67 labeling index. The WHO classification system distinguishes well-differentiated tumors as NENs while poorly differentiated tumors are referred to as neuroendocrine carcinoma (NEC). [4]. Grade (G)1 NENs have a mitotic count <2 per 10 high powered fields (HPF) and a Ki-67 index < 3% while G2 NENs have a mitotic count 2–20 per 10 HPF and/or Ki-67 index 3–20%. G3 NENs display a well-differentiated morphology with a mitotic count >20 per 10 HPF and/or a Ki-67 index > 20% while G3 NECs display poorly differentiated morphology with a mitotic count >20 per 10 HPF and/or a Ki-67 index > 20%.

Treatment options for G1 and G2 NENs consist of surgery (primary and cytoreductive), somatostatin analogs (SSAs), liver directed therapy (bland embolization, chemoembolization, radioembolization), radionuclide therapy (Lutetium-177 [^177^Lu]-DOTATATE, Iodine-131 metaiodobenzylguanidine [^131^I-MIBG]), targeted therapy (everolimus, sunitinib and interferon), or chemotherapy [5,6,7]. G3 NENs with well-differentiated morphology (predominantly those with a Ki-67 index between 21–54) have been shown to respond well to treatments geared towards G1 and G2 NENs [5,8]. G3 NECs generally do not respond to the standard treatments traditionally utilized in G1 and G2 NENs [5,9,10]. Multiple studies have shown the efficacy of platinum-based chemotherapy in G3 NECs [11,12,13,14]. As such, the National Comprehensive Cancer Network (NCCN) guidelines currently recommend first-line treatment for G3 NECs to consist of etoposide and cisplatin or carboplatin [5].

The role of chemotherapy in NENs has continued to improve and evolve. Yet, the best chemotherapy agent to use in G3 NECs has yet to be determined. Multiple studies have shown promising results in treating NENs using a regimen combining capecitabine and temozolomide (CAPTEM). This therapy has revealed a response rate of up to 70% in G1 and G2 NENs [15,16,17,18,19,20]. Additionally, a recent prospective study demonstrated that CAPTEM was associated with an improved progression free survival (PFS) and overall survival (OS) in G1 and G2 NENs in metastatic pancreatic NENs (pNENs) when compared to temozolomide alone [21]. Data supporting the use of CAPTEM in non-pNENs as well as G3 NENs and G3 NECs is scarce. Eads and colleagues are currently conducting the first prospective study to investigate the use of CAPTEM in the treatment of G3 NECs compared to platinum-based chemotherapy; however, its efficacy remains unclear (NCT02595424).

The New Orleans Louisiana Neuroendocrine Tumor Specialists (NOLANETS) are a multidisciplinary team formed as a collaborative effort between Ochsner Medical Center in Kenner, Louisiana and Louisiana State University Health Sciences Center in New Orleans, Louisiana. The NOLANETS conducted a retrospective study in 2016 that examined 29 patients with NENs who received at least one cycle of CAPTEM [22]. The results of this study suggested that CAPTEM may potentially be beneficial as a treatment option for these patients. Since the time of these reported findings, 80 additional patients were treated with CAPTEM at our institution. Thus, the purpose of this study was to build on the results of our previously mentioned report [22]. Specifically, the aim of this study was to perform an evaluation of the outcome and role for CAPTEM in malignant pNENs and non-pNENs. To the best of our knowledge, the current study is the largest series to include patients with non-pNENs.

## 2. Methods

This study was conducted in accordance with the Declaration of Helsinki, and the protocol was approved by the Institutional Review Boards at Louisiana State University Health Sciences Center-New Orleans and Ochsner Clinic Foundation (project identification code: #7723). A waiver of informed consent from the IRB was received. Our study was a retrospective chart review, in which pre-recorded patient data were gathered from their electronic medical record. Clinical data from all patients with histopathologic diagnosis of NEN who were evaluated by the NOLANETS team were entered into a web-based patient database (VELOS Inc., Fremont, CA, USA). The database was queried for patients who were treated with CAPTEM from November 2010 to June 2018 who sought care at the NOLANETS Clinic. Only patients who received at least one cycle of CAPTEM were included. Patients who had incomplete records or were lost to follow-up were excluded from the study. We defined lost to follow-up as a patient who had only one visit in our clinic without evidence that their treatment plan was initiated or had no follow up imaging performed. Inclusion and exclusion criteria are shown in Figure 1.

Demographics (age, sex, race/ethnicity, number of prior treatments, and metastatic disease), pathologic characteristics (Ki-67 proliferative rate, tumor differentiation, and WHO grade), imaging results (CT scans, Oscan, MIBG, PET), as well as previous and current treatment data were collected from our database and supplemented by medical record review. A grade was assigned to each patient during chart review based on Ki-67 and mitotic index according to “principles of pathology for the diagnosis and reporting of neuroendocrine tumors” in the NCCN guidelines, version 1.2019 [5]. Patients received capecitabine 750 mg/m^2^ by mouth twice daily on days 1–14 and temozolomide 200 mg/m^2^ by mouth once daily on days 10–14. Each cycle was 28 days long.

Baseline survival was defined as 16.2 years, 8.3 years, and 10 months for G1, G2, and G3 NENs, respectively [23]. Our primary outcome was progression free survival (PFS) defined as time from the first CAPTEM treatment until disease progression or death. Secondary outcomes consisted of overall survival (OS), overall response rate (ORR) and disease control rate (DCR). OS was defined as time from the first CAPTEM treatment to death or the termination of the study (June 1, 2018). Death due to all causes was used as the outcome parameter. Patients who were alive at the end of the study were treated as censored. ORR was defined as the proportion of patients who had a partial response (PR) or complete response (CR) to therapy according to Response evaluation criteria in solid tumors (RECIST) criteria. DCR was defined as the proportion of patients who had stable disease (SD), PR or CR according to RECIST criteria.

Patients were evaluated every cycle with blood work (complete blood count and comprehensive metabolic panel) to monitor for toxicities and were assessed radiographically every three cycles. Response to treatment was assessed on CT and/or MRI images according to response evaluation criteria in solid tumors (RECIST 1.1) parameters. Treatment-related adverse events were graded according to the common terminology criteria for adverse events (CTCAE v.5.0).

Strategies to reduce bias and enhance the validity, reproducibility and quality of the chart review were implemented. These included defining variables, testing reliability between data miners, and research meetings to discuss discrepancies or issues that arose during chart reviews.

Statistical analyses were performed using MedCalc Statistical Software version 18.10.2 (MedCalc Software), Ostend, Belgium; http://www.medcalc.org; 2018). Descriptive statistics were used to summarize patient characteristics. Chi-square tests were used to test for associations between patient characteristics and ORR or clinical benefit. Survival curves for PFS and OS were estimated using the Kaplan-Meier method and compared using the log-rank test. PFS was reported as 1-year and median survival rates, while OS was reported as 2-year and median survival rates. Median PFS and OS are expressed with 95% confidence intervals (CIs). Subset analyses were performed to evaluate PFS, OS, ORR and DCR stratified by Ki-67 proliferative rate and primary tumor site. *p*-values < 0.05 were considered statistically significant.

Cox proportional hazards models were used to further identify individual factors associated with prognosis. All factors that were statistically significant in the univariate analysis (*p* < 0.05) were included as covariates in multivariable analyses. Unadjusted and adjusted hazard ratios are presented with 95% confidence intervals.

## 3. Results

We identified 116 patients with metastatic NENs who received at least one cycle of CAPTEM between 1 November 2010 and 1 June 2018. Primary and metastatic site was identified in all 116 patients. Ki-67 proliferative rate was recorded in 106 patients. Median age at diagnosis was 56 years (range: 17–83). Median time from initial diagnosis to initiation of CAPTEM therapy was 18.5 months (Range: 0–265). The median number of cycles of CAPTEM was 9.5. Table 1 summarizes the demographics and characteristics of our cohort.

Using RECIST parameters, the DCR and ORR for our entire cohort was 73% (*n* = 85) and 21% (*n* = 24). Table 2 summarizes the response to CAPTEM treatment using RECIST parameters by tumor characteristics and prior treatments in our cohort. ORR and DCR were calculated for patients who underwent prior surgery, SSA therapy, chemotherapy, targeted therapy, and peptide receptor radionucleotide therapy (PRRT).

CAPTEM treatment was discontinued in 89 patients (77%) in our study. Of those who discontinued CAPTEM, 41 patients (46%) stopped due to progression, 22 patients (25%) due to an adverse event, and 22 patients (25%) due to the decision of patient or treating physician. In 4 patients (4%), CAPTEM treatment was discontinued due to patient death that was unrelated to an adverse event. Adverse events during CAPTEM therapy were observed in 74 patients (64%). The most common toxicities observed were nausea/vomiting in 30 patients (26%) and fatigue in 26 patients (22%). CAPTEM treatment toxicities stratified by Common Terminology Criteria for Adverse Events (CTCAE) v5.0 Grade are shown in Table 3. Of the 36 patients who had previous chemotherapy, 11 (30.5%) reported grade 3 or grade 4 toxicity. Among those who did not have previous chemotherapy, 13 (16.3%) reported a grade 3 or 4 toxicity.

Forty-nine patients (42%) died by the end of this study. Median PFS was 13 months (95% CI: 11–23). Kaplan-Meier 1-year PFS was 51% (Figure 2a). Median OS was 38 months (95% CI: 32–46). Kaplan-Meier 2-year OS rate was 69% (Figure 2b). Survival analysis after CAPTEM treatment sorted by tumor characteristics is shown in Table 4.

PFS stratified by pNENs vs. non-pNENs was statistically significant (log-rank *p* = 0.0083). Median PFS for pNENs was 29 months (95% CI: 12–80) compared to 11 months (95% CI: 6–18) in non-pNENs. Kaplan-Meier 1-year PFS rate for pNENs was 61% compared to 44% for non-pNENs (Figure 3a). OS stratified by pNENs vs. non-pNENs was not statistically significant (log-rank *p* = 0.5621). Median OS for pNENs was 35 months (95% CI: 29–80) compared to 38 months (95% CI: 30–46) in non-pNENs. Kaplan-Meier 2-year OS rate for pNENs was 74% compared to 66% for non-pNENs (Figure 3b).

Median PFS was 29 months in patients with a Ki-67 < 3% (95% CI: 13–39), 12 months in patients with a Ki-67 = 3–20% (95% CI: 6–24), 7 months in patients with a Ki-67 > 20–54%, (95% CI: 3–25), and 5 months in patients with a Ki-67 ≥ 55% (95% CI: 4–12). These results were statistically significant (0.0287). The Kaplan-Meier 12-month and 24-month PFS for Ki-67 < 3%, was 70% and 60%, Ki-67 = 3–20% was 45% and 35%, Ki-67 > 20–54% was 41% and 23%, and Ki-67 ≥ 55% was 17% and 0%, respectively, estimated from the time of initial treatment with CAPTEM (Figure 4a).

Median OS was not reached among those with a Ki-67 < 3%, 33 months in patients with a Ki-67 = 3–20% (95% CI: 30 to 46), 33 months in patients with a Ki-67 > 20–54%, (95% CI: 14–46), and 18 months in patients with a Ki-67 ≥ 55% (95% CI: 15–25). Statistical significance was not reached (*p* = 0.0779). The Kaplan-Meier 2-year and 5-year OS for Ki-67 < 3%, was 79% and 51%, Ki-67 = 3–20% was 70% and 28%, Ki-67 > 20–54% was 61% and 22%, and Ki-67 ≥ 55% was 33% and 0%, respectively, estimated from the time of initial treatment with CAPTEM (Figure 4b).

Factors significantly associated with PFS according to the univariate Cox models were pNEN status (*p* = 0.012), tumor differentiation (*p* = 0.026), WHO grade G3 NEN/G3 NEC (*p* = 0.023), and Ki-67 proliferative rate > 20% (*p* = 0.013). After adjusting for all of these factors in the multivariate model, only pNEN status was associated with PFS. Specifically, after adjustment, patients with pNEN had a 50% lower hazard of disease progression compared to those with non-pNEN (adjusted HR: 0.498, 95% CI: 0.29–0.85, *p* = 0.0100) (Table 5).

## 4. Discussion

Overall, the present study shows that CAPTEM was beneficial in NENs of various tumor proliferation rates and locations. In our study, 73% of patients demonstrated a disease control to using CAPTEM, indicated by complete, partial response or stable disease. Additionally, patients experienced an overall median PFS of 13 months and overall survival of 38 months.

Our patients also experienced a high rate of adverse effects, with more than half reporting grades 1, 2, 3, or 4 toxicity. In fact, having previous chemotherapy almost doubled the incidence of grade 3 or gastrointestinal in nature. Perhaps patients were better primed to report these symptoms, having previously gone through chemotherapy regimens and being aware of potential toxicities.

Other retrospective studies show similar results. Owens (2017) reports a PFS of 13 months and OS of 29.3 months among patients with NEN receiving CAPTEM, regardless of site or grade. Additionally, the DCR was 90% of all patients undergoing this treatment regimen [24].

Our pNEN patients had significantly better PFS on CAPTEM, when compared to non-pNEN, even after adjusting for tumor differentiation and Ki-67 proliferative rate. This finding is not novel; the trend for pNENs to respond better than non-pNEN to CAPTEM been reported throughout the literature [24,25]. The reason for this remains unknown. One thought is that capecitabine may reduce O6 methylguanine DNA methyltransferase (MGMT), a DNA repair enzyme, causing the cell to be more vulnerable to the properties of temozolomide [26]. MGMT deficiency has been found to be more prominent in pNENs (51%), when compared to non-pNENs (0%) [27]. This inherent deficiency in pNENs may account for the discrepancy between tumor site efficacy. Ultimately more research is needed to determine why pNENs respond better to chemotherapy than non-pNENs. In an ongoing Phase II clinical trial, Fine et al. (2014) examined the effect of CAPTEM on patients with NEN who progressed on high dose octreotide (NCT00869050) [17]. An interim analysis of this study reported that PFS in pNEN is greater than 18 months, which is 150% greater than the response achieved with targeted therapies (everolimus and sunitinib) [17]. Additionally, Owen et al. (2017) found that pNEN had a significantly longer PFS of 16.7 months, when compared to 8.4 months in non-pNEN patients [24]. The results from these studies are in accordance with our data; in fact, our pNEN patients demonstrated an even longer PFS than reported in these studies, achieving 29 months before progressing on CAPTEM. Moreover, unlike our study that included NENs of all differentiation, Fine et al. included only metastatic well-moderately differentiated NEN (Ki-67 ≤ 20%) [17]. This makes the findings in our study even more thought-provoking, as one might expect that the well differentiated tumors with a lower Ki-67 value in Fine and colleagues’ study to do better overall. We should note, however, that our study included over four-fold the number of patients, which may account for variations in data findings. Our study population achieved a PFS among non-pNENs that was considerably longer than findings reported in other studies, albeit much less than PFS in pNENs. Peixoto et al. (2014) demonstrated a median PFS of only 2.8 months, which is pointedly lower than the 11 months reported in our study [25].

Despite these findings, the OS between pNENs and non-pNENs was not statistically significant. However, given that the OS varied by five months, this may be of considerable clinical importance. ORR was found to be 38% and 9% for pNENs and non-pNENs, respectively. Interestingly, these data are significantly lower than found in the literature. Strosberg et al (2011) reported a 70% radiographic response rate with metastatic pNENs treated with first line CAPTEM [20]. The reason for this may be twofold. The first may be due to the fact that our study population was heavily pretreated, with 91% (*n* = 105) of the entire cohort of patients having prior cytoreduction, targeted therapy, radionuclide therapy, liver-directed or chemotherapy, or a combination of these treatments. Secondly, this discrepancy may be owing to the relatively small sample size of this study (*n* = 30), as one alteration in clinical course could skew the results and artificially increase the response rate.

Despite the low response rate reported in this manuscript, our PFS among patients with pNENs was high and our results are in line with the current literature. Kunz et al. (2018) conducted a randomized, phase 2 trial comparing temozolomide verses CAPTEM in 144 patients with pNENs [21]. The median PFS was 22.7 months for CAPTEM (versus 14.4 months for those treated with temozolomide alone). Kunz et al. results demonstrated the longest PFS reported among patients with pNENs. Our current study, albeit not a randomized control trial, demonstrated an even longer PFS of 29 months; these results lend support for the use of CAPTEM for pNEN directed therapy [21].

This is the first study to examine the effect of a CAPTEM regimen on PFS among four distinct groups based on Ki-67. In line with the common belief that cytotoxic therapy works better among poorly differentiated tumors, our analysis showed that those with a higher Ki-67 level had a significantly worse PFS compared to those with a lower Ki-67 value [28]. Furthermore, we hypothesized that Ki-67 would correspond with PFS, although no statistical significance was achieved after adjusting for other factors, a trend towards improved PFS was seen in those with a lower Ki-67. While our study has one of the largest sample sizes to date among studies looking at CAPTEM regimens in NEN, examining only 108 patients of varying Ki-67 values may have limited statistical power.

Furthermore, the benefit seen in patients with G3 NEN and Ki-67 ranging from >20–54% merits further clinical research. The NCCN guidelines state that NENs with Ki-67 > 20–50% with well differentiated histology may respond comparatively poorly to chemotherapy but do better with treatment regimens designed for G1 and G2 tumors [5]. Our study challenges this idea, as 58% of patients with a Ki-67 > 20–54% achieved a clinical benefit with this regimen (compared to 50% with Ki-67 ≥ 55), indicating that CAPTEM may be beneficial for tumors with a high Ki-67 and well differentiated histology. Sahu and colleges (2019) also found that there was a trend towards increased PFS in those with a Ki-67 20–54%, compared to patients with a Ki-67 ≥ 55% (15 vs. four months, *p* = 0.11) [29]. This lends further support for additional examination of CAPTEM in these patients.

In the past year, recent studies have reported on the use of CAPTEM in advanced NEN [29,30,31,32,33]. Chatzellis et al. (2019) conducted a retrospective analysis of CAPTEM in 79 patients, of which 30 patients (38%) had pNEN and 15 patients had GI NEN primary disease. The median PFS and OS was 10.1 and 102.9 months, respectively. Using multivariable analysis, Chatzellis et al. found that pancreatic or lung tumor primary site were an independent prognostic factor for PFS (*p* = 0.002) and OS (*p* = 0.028) [30].

Additionally, de Mestier and colleagues (2019) compared CAPTEM with 5-fluorouracil (5FU)-dacarbazine (DTIC) in 247 patients with NEN. It was found that PFS was similar between the two groups 13.9 vs. 18.3 months, *p* = 0.86) [31]. Parallel to Chatzellis’ study, the majority of patients getting CAPTEM had a pNEN primary disease (82.3%), and only 27 patients with non-pNENs. In this study, the use of CAPTEM in pNET was associated with a significantly higher ORR, PFS, and OS when compared to patients with non-pNENs [30,31].

This highlights the fact that pNENs tend to be more chemosensitive, when compared to non-pNENs (deMestier, 2019). As such, these patients generally have a more favorable response to CAPTEM treatment [19].

The use of the CAPTEM regimen shown significant activity in pNENs but the role for this treatment in non-pNEN has yet to be established. [34,35] Additionally, although many of our patients reported a treatment associated toxicity, the literature states that CAPTEM is a relatively well tolerated treatment option for patients with NENs. Chatzellis et al. (2019) report that CAPTEM treatment is rarely associated with serious toxicities and is associated with low discontinuation rates, even in patients who are on the treatment for more than one year [30].

Lacking in the current literature is the investigation of CAPTEM in a larger population of non-pNENs. Recently published manuscripts have investigated the use of CAPTEM in this population. However, the number of patients available for study remain low [29,32,33,36].

To the best of our knowledge, our current study is the largest series to include patients with non-pNEN. Our study builds on the current knowledge by describing the use of CAPTEM in 69 patients with non-pNEN.

## 5. Limitations

One of the most apparent limitations to this study is its retrospective design and potential for selection bias from chart review; however, given the rarity of this tumor, prospective studies are difficult to attain. Data from these small, nonrandomized studies make it challenging to draw specific treatment recommendations. To date, this report is one of the largest to look at the effect of a CAPTEM regimen across the broad spectrum of NENs.

The Louisiana State University/Ochsner Neuroendocrine Tumor Program is a specialty center and the referral base that incorporates both national and international regions. Patients who come to the clinic from great distances are characteristically co-managed with their local, primary oncologist. This may limit the accessibility of sequential radiologic data. However, if patients did not have routine follow-up in our clinic or adequate outside records, they were excluded from this study.

Our study is also limited by the potential survivor treatment selection bias. Additionally, our population is heterogenous and would ideally have a matched control group. Also, we did not collect post treatment data and, as such, we aren’t able to determine if the differences in OS might be affected by therapies administered after progression. Presumably, many patients went onto treatment with Lu-177 since the United States Food and Drug Administration approval was in January 2018. Lastly, as our specialty center obtained patients from various medical practices, initial histological and radiological reports were generated by a variety of physicians outside of our treatment center. As such, our study lacks centralized histological and radiological assessment.

## 6. Conclusions

This study describes our findings with the use of a CAPTEM regimen in a large number of NEN patients with varying Ki-67 and tumor location. Our results indicate that CAPTEM is associated with encouraging PFS, OS, DCR, and objective response data in patients with NEN. Our data supports previous retrospective studies for the use of CAPTEM in pNENs. Additionally, this study adds to the limited knowledge base of CAPTEM use in non-pNENs and high grade NENs and advocates for the potential role of this therapy in these subgroups. Further research is needed to make definite recommendations regarding the use of this chemotherapy option in NENs of varying Ki-67 values and locations.

## Figures and Tables

**Figure 1 cancers-12-00206-f001:**
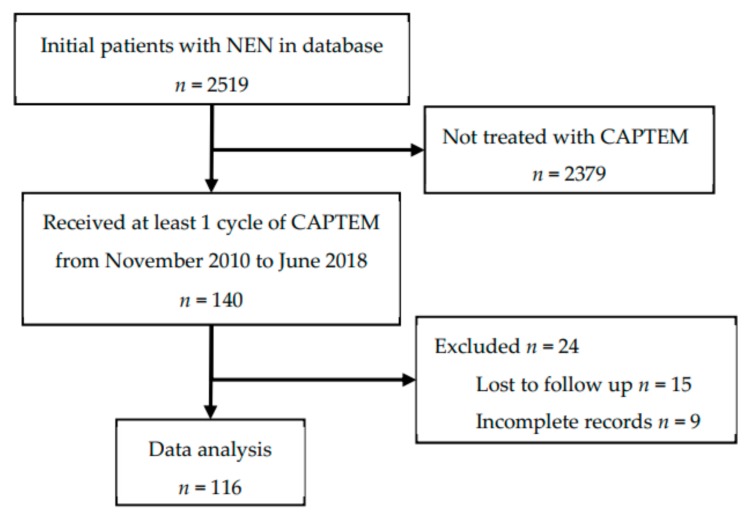
Inclusion and exclusion criteria.

**Figure 2 cancers-12-00206-f002:**
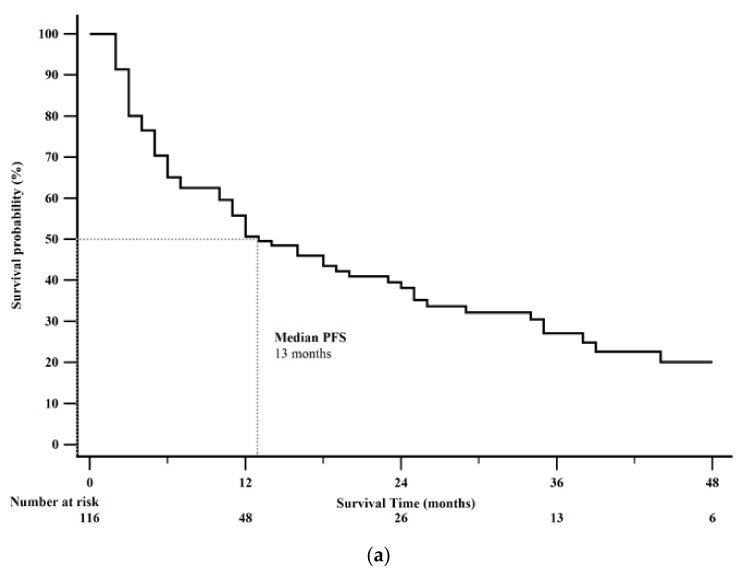
Kaplan-Meier survival curves from the date of initiation of Capecitabine and Temozolomide (CAPTEM) for the entire cohort (*n* = 116). (**a**) Median progression free survival (PFS) was 13 months (95% CI: 11–24). Kaplan-Meier 1-year progression free survival rate was 51%. (**b**) Median overall survival was 38 months (95% CI: 32–46). Kaplan-Meier 2-year overall survival rate was 69%.

**Figure 3 cancers-12-00206-f003:**
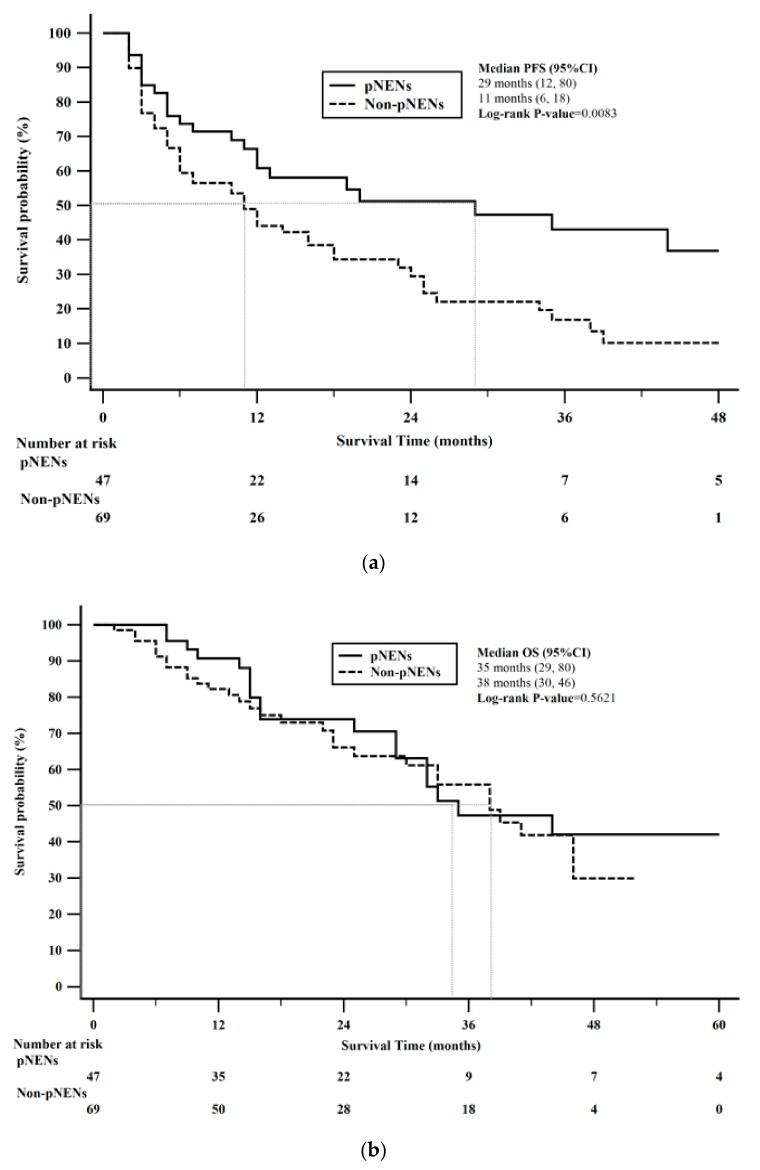
Kaplan-Meier survival stratified by pancreatic neuroendocrine neoplasms (pNENs) versus non-pancreatic neuroendocrine neoplams (non-pNENs). (**a**) Progression free survival (PFS) stratified by pNENs and non-PNENs was statistically significant (*p*-value = 0.0083). (**b**) Overall survival (OS) stratified by pNENs and non-PNENs was not statistically significant (*p*-value = 0.5621).

**Figure 4 cancers-12-00206-f004:**
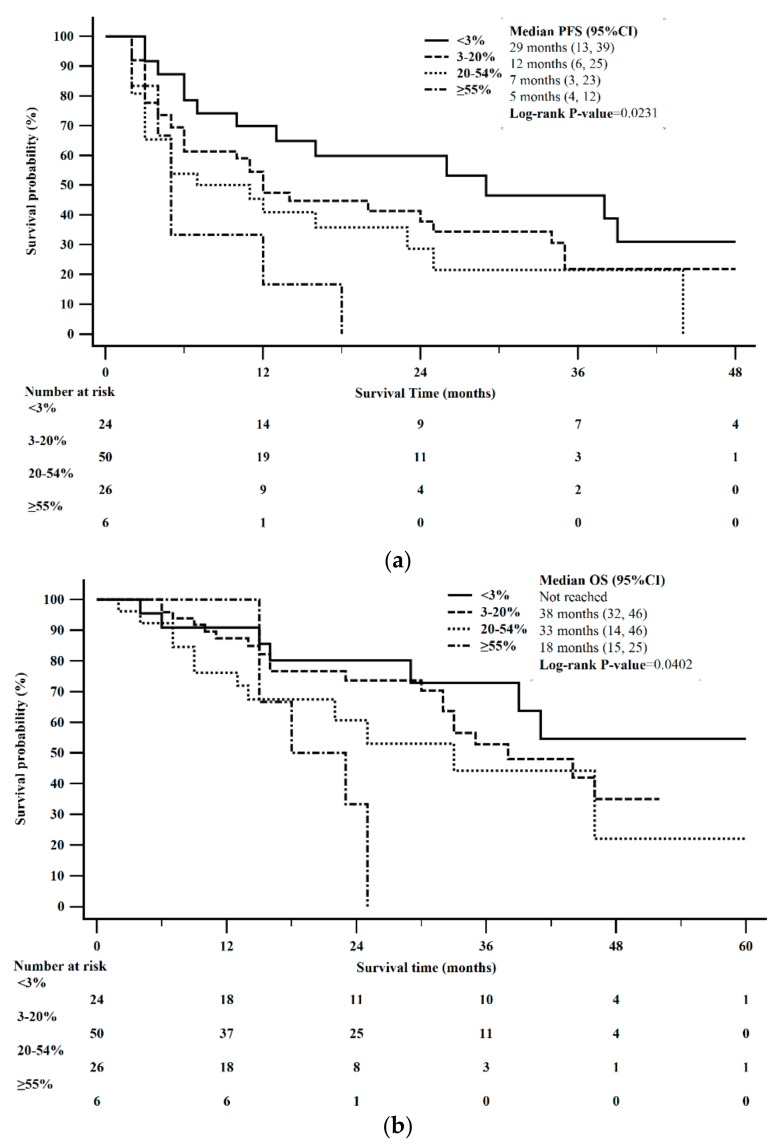
Kaplan-Meier survival stratified by Ki-67. (**a**) PFS differed between Ki-67 proliferative rate (log-rank *p* = 0.0231). (**b**) OS was not significantly different between Ki-67 proliferative rate (log-rank *p* = 0.0402).

**Table 1 cancers-12-00206-t001:** Characteristics of the Study Cohort (*n* = 116).

Characteristic	*n* (%)
*Sex*	
Female	48 (41)
Male	68 (59)
*Tumor Differentiation*	
Poor	18 (16)
Well	94 (81)
Unknown	4 (3)
*Primary Tumor Site*	
Pancreas	47 (41)
Small Intestines	37 (32)
Unknown Primary Site	12 (10)
Lung	12 (10)
Colon/Rectum	7 (6)
Kidney	1 (1)
*Metastatic Disease*	
None	3 (3)
Regional Lymph Node	4 (3)
Hepatic only	59 (51)
Extrahepatic only	6 (5)
Hepatic & Extrahepatic	44 (38)
*Prior Treatment*	
None	11 (9)
Somatostatin Analogs	54 (47)
Surgery	67 (58)
Chemotherapy	36 (31)
Liver Directed Therapy	35 (30)
Targeted Therapy	19 (16)
Radionuclide therapy	4 (4)
Radiation Therapy	4 (4)
*Treatment Lines Prior to CAPTEM*	
1	34 (30)
2	43 (37)
3	22 (19)
4	12 (10)
5	5 (4)

Shown is the frequent of patients who received prior treatment per treatment modality. Abbreviation: CAPTEM, capecitabine and temozolomide.

**Table 2 cancers-12-00206-t002:** Response to CAPTEM treatment using RECIST parameters sorted by Tumor Characteristic.

Tumor Characteristics	Radiographic Response, *n* (%)					ORR		DCR	
	*n*	CR	PR	SD	PD	*n* (%)	*p* Value	*n* (%)	*p* Value
All patients	116	1 (1)	23 (20)	61 (53)	31 (27)	24 (21)		85 (73)	
Primary Site (*n* = 116)									
pNEN	47	1 (2)	17 (36)	18 (38)	11 (23)	18 (38)	*0.0001*	36 (77)	0.5049
Non-pNEN	69	-	6 (9)	43 (62)	20 (29)	6 (9)		49 (71)	
Ki-67 (*n* = 106) *									
Ki-67 < 3%	24	-	5 (21)	17 (71)	2 (8)	5 (21)	0.5241	22 (92)	*0.0084*
Ki-67 3–20%	50	1 (2)	12 (24)	24 (48)	13 (26)	13 (26)		37 (74)	
Ki-67 20–55%	26	-	3 (12)	12 (46)	11 (42)	3 (12)		15 (58)	
Ki-67 > 55%	6	-	1 (17)	1 (17)	3 (50)	1 (17)		2 (33)	
Prior Treatments (*n* = 109)									
No SSA	58	0 (0)	12 (21)	32 (55)	14 (24)	12 (21)	0.6877	44 (76)	0.6928
SSA	51	1 (2)	8 (16)	28 (55)	14 (27)	9 (18)		37 (73)	
No Surgery	47	0 (0)	11 (23)	22 (47)	15 (32)	11 (23)	0.6487	33 (70)	0.2195
Surgery	62	1 (2)	11 (18)	37 (60)	13 (21)	12 (19)		49 (79)	
No Chemotherapy	75	1 (1)	14 (19)	45 (60)	15 (20)	15 (20)	0.6020	60 (80)	0.0666
Chemotherapy	34	0 (0)	6 (18)	17 (50)	1 (3)	6 (18)		23 (68)	
No Targeted Therapy	91	1 (1)	16 (18)	52 (57)	22 (24)	17 (19)	0.8400	69 (76)	0.7463
Targeted Therapy	18	0 (0)	3 (17)	10 (56)	5 (28)	3 (17)		13 (72)	
No PRRT	106	1 (1)	20 (19)	58 (55)	27 (25)	21 (19)	0.3909	79 (75)	0.3135
PRRT	3	0 (0)	0 (0)	3 (100)	0 (0)	0 (0)		3 (100)	

Shown is the frequency of each type of response per RECIST 1.1 per tumor characteristic and prior treatment. Abbreviations: CAPTEM, capecitabine and temozolomide; SSA, somatostatin analog; CR, complete response; PR, partial response; SD, stable disease; PD, progressive disease; ORR, overall response rate; DCR, disease control rate; PRRT, peptide receptor radionucleotide therapy. * Ki-67 unknown (*n* = 10).

**Table 3 cancers-12-00206-t003:** CAPTEM treatment toxicities stratified by common terminology criteria for adverse events (CTCAE) v5.0 Grade (*n* = 116).

Toxicity	All Grades, *n* (%)	Grade 1	Grade 2	Grade 3	Grade 4
Anemia	8 (7)	-	3	5	-
Thrombocytopenia	11 (9)	-	2	6	3
Lymphopenia	8 (7)	-	3	5	-
Neutropenia	6 (5)	-	1	4	1
Fatigue	26 (22)	7	12	6	1
Nausea/Vomiting	30 (26)	10	14	5	1
Diarrhea	10 (9)	3	3	2	2
PPE	13 (11)	1	4	7	1
Weight loss	3 (3)	3	-	-	-
Other	14 (12)	7	2	3	2

Shown here is the frequency of each specific toxicity per grade. Abbreviation: CAPTEM, capecitabine and temozolomide; CTCAE, Common Terminology Criteria for Adverse Events, PPE, Palmar-plantar erythrodysesthesia.

**Table 4 cancers-12-00206-t004:** Survival analysis after CAPTEM treatment sorted by Tumor Characteristics.

Factors	*n*	Median PFS (95% CI)	*p* Value (Log-Rank)	Median OS (95% CI)	*p* Value (Log-Rank)
All patients	116	13 (11–23)		38 (32–46)	
Sex			0.1713		0.1166
Female	48	25 (10–35)		NR	
Male	68	12 (7–18)		33 (29–41)	
Primary Site (*n* = 116)			0.0083		0.5621
pNEN	47	29 (12–80)		35 (29–80)	
Non-pNEN	69	11 (6–18)		38 (30–46)	
Tumor Differentiation (*n* = 112)			0.0192		0.0192
Well	94	16 (11–26)		41 (33–46)	
Poor	18	5 (5–12)		23 (13–25)	
Ki-67 (*n* = 106) *			0.0231		0.0402
Ki-67 < 3%	24	29 (13–39)		NR	
Ki-67 3–20%	50	12 (6–25)		33 (30–46)	
Ki-67 20–55%	26	7 (3–25)		33 (14–46)	
Ki-67 > 55%	6	5 (4–12)		18 (15–25)	

* Ki-67 unknown (*n* = 10). Abbreviation: CAPTEM, capecitabine and temozolomide; PFS, progression free survival; OS, overall survival.

**Table 5 cancers-12-00206-t005:** Proportional Hazards Models for PFS by Tumor Characteristics after CAPTEM treatment.

Tumor Characteristics	Univariate Analysis		Multivariate Analysis ^†^	
	HR (95% CI)	*p* Value	HR (95% CI)	*p* Value
Primary Site				
PNEN	1		1	
Non-PNEN	1.89 (1.15–3.11)	0.0116	2.04 (1.20–3.47)	*0.0088*
Tumor Differentiation				
Well	1		1	
Poor	1.92 (1.08–3.41)	0.0263	1.07 (0.48–2.37)	0.8682
Ki-67 *				
Ki-67 < 3%	1		1	
Ki-67 3–20%	1.62 (0.84–3.09)	0.1477	11.35 (2.42–53.29)	*0.0021*
Ki-67 > 20%	2.36 (1.20–4.65)	0.0129	7.76 (0.60–100.50)	0.1167

* Ki-67 unknown (*n* = 10). ^†^ Adjusted for all tumor characteristics in the table. Abbreviations: CAPTEM, capecitabine and temozolomide.

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
