# Peer review of "Outcomes of Capecitabine and Temozolomide (CAPTEM) in Advanced Neuroendocrine Neoplasms (NENs)"

_cancers, 2020, doi:10.3390/cancers12010206_

Round 1

Reviewer 1 Report

This is an interesting paper in a large "real-world" cohort of patients with NETs G1- G3. However, a high rate of patients is rated as NET G3, although this entity has aonly been defined in the ENETS 2017 guiodeoines for pNET, and only now been adopted for non-pancreatic origin. In view of this, the authors might include a short paragraph on NET G3 and the high rate in their series, which spans a period where only in the last part official Definition had been done?

The findings are of interest to the oncologic community, and also the fact that pNETs are different to non-pNETs deserve publication. However, the differences in OS might be influenced by therapies administered after progression (or also in patients who might have shown only stable disease, but were symptomatic which might also have influenced PFS), so therapies given after CAPTEM should be included in the paper.

Author Response

REVIEWER 1

1. This is an interesting paper in a large "real-world" cohort of patients with NETs G1- G3. However, a high rate of patients is rated as NET G3, although this entity has a only been defined in the ENETS 2017 guidelines for pNET, and only now been adopted for non-pancreatic origin. In view of this, the authors might include a short paragraph on NET G3 and the high rate in their series, which spans a period where only in the last part official Definition had been done?

Responses: This has been addressed in the introduction.

References have been added.

2. The findings are of interest to the oncologic community, and also the fact that pNETs are different to non-pNETs deserve publication. However, the differences in OS might be influenced by therapies administered after progression (or also in patients who might have shown only stable disease, but were symptomatic which might also have influenced PFS), so therapies given after CAPTEM should be included in the paper.

Responses: In this retrospective chart review, we did not collect post treatment data. Presumably many patients went onto receive other treatment (specifically, PRRT).  This has been added as a limitation in the study.

Reviewer 2 Report

The authors present a retrospective analysis of the use of CAPTEM regimen in a heterogeneous group of patients with NENs validating previous studies in this field. The data are of interest to the clinical community due to the largest so far cohort with real-world data, however there are some flaws that need to be addressed. The manuscript would benefit from further revision and clarification with respect to its limitations. In particular, subgroup analyses should be performed defining NEN population characteristics treated with CAPTEM more consistent with clinical practice.
Abstract
1. Define all abbreviations in the abstract: i.e.NEN, PFS, OS, ORR, pNEN, NEC, G1, G2.
2. Comprehensive data are needed on the duration of CAPTEM study treatment in this study. There is currently a great controversy and a lack of evidence on fixed duration CAPTEM regimens vs. prolonged treatment until progression or toxicity occurrence. Please explain in the methods part of the abstract.
3. Safety analysis of CAPTEM should be added.
4. Please rephrase the conclusions and consider presentation of CAPTEM safety analysis. The sentence” The current study is the largest series to include 35 patients with non-pNENs” is not a study conclusion. It rather belongs to the discussion section.
Introduction
This part is well-written and provides an overview in the field of available treatments including chemotherapy, and CAPTEM in particular, in NENs.
Methods
1. Page 3, line 108 and 110. Please provide references in right format.
2. Please provide staging and grading system for different NEN primaries and relevant references in the methods section. Please explain how you managed patients grading before 2017, as the NEN G3 entity was introduces that year by the WHO classification for pNENs.
3. Did the study include central pathology review?
4. Please define baseline for survival estimates.
5. As stable disease may be a favorable outcome particularly in NEN patients who received previous multiple lines of other treatments and progressed, please consider to provide data on disease control rate (DCR).
6. Please use STROBE to report the findings of the analysis. Provide relevant analysis.
Results
1. Please include descriptive data in the first two paragraph of the results’ section (page 4, lines 137-148) in Table 1 instead. The second paragraph is already a duplication.
2. Please specify number of lines of treatment prior to CAPTEM administration and include this information in Table 1.
3. Page 5, section 3.2 on Treatment Response is very difficult to follow. Please use STROBE to report your results. This data should be mainly presented only in Table 2. WHO grade and Ki67 subgroup analysis are overlapping and one should be omitted as specified below (please see comments for tables).
4. Please provide data on ORR and DCR also for the following strata: functionality (secretory status), FDG/Galium68-PET avidity, previous surgery, different treatments and specifically previous SSA, chemotherapy, PRRT and molecular targeted therapies.
5. Please provide date on toxicities over time (when do they occur?) and overall discontinuation rates due to toxicities. Does previous chemotherapy increases the risk for serious toxicities with CAPTEM?
6. OS and PFS analysis of non-pNENs altogether is not comprehensive as this is a heterogenous group and colon NENs and UPOs are for example much more aggressive than SI-NENS and lung NENS. Please consider providing KM curves and log-rank analysis for each primary in two figures, one for PFS and one for OS.
7. Same applies to log-rank analysis and relevant figures for grade. Please use instead Ki67 index grouping and add the cut-off of 55% within G3 to address the difference in G3NEN vs. G3NEC (i.e 4 Ki67 groups)
8. Multivariable Cox regression analysis should omit overlapping parameters and potentially include factors that I previously suggested to test in univariable analysis, i.e. functionality (secretory status), FDG/Galium68-PET avidity, previous surgery, different treatments and specifically previous SSA, chemotherapy, PRRT and molecular targeted therapies, number of previous lines of treatments.
Discussion.
1. First paragraph in discussion should only present and elaborate on the study’s own results without making comparisons with other studies at this point. Safety analysis considerations and length of treatment aspects could be briefly discussed here.
2. OS analysis conclusions are hard to make as these patients have received different line of treatments that preceded or followed CAPTEM. As such, the discussion in lines 283-284 is hard to comprehend.
3. Line 301: The authors provide analysis results in 3 Ki67 categories and have put together G3NENs and NECs, so this claim is not true. Please consider providing complementary analysis to support these claims.
4. Safety analysis should be discussed with respect to studies on CAPTEM that the authors refer to.
5. Limitations: The study is limited by the potential selection bias toward long survivors and the heterogeneous population of patients without well-matched subgroups and control groups. Centralized radiological and histological assessment?
6. Please specify future aspects that need to be addressed in conclusions.
Tables and Figures
1. Please explain why you present both grade and Ki67 proliferation index subgroup analysis in Table 2. Why is there a discrepancy in these groups? I would suggest to omit the grade grouping and use only the Ki67 one (add an extra subgroup there, i.e. ki67 20-55 and ki67>55). Lung NENs have other Ki67 cut-offs and other parameters such as type (typical/atypical) are more relevant in terms of prognosis. Using both grade and Ki67 grouping is actually confusing.
2. Please apart from ORR, include DCR analysis in Table 2, as discussed above.
3. Table 4: In univariable log-rank analysis the parameters selected are party mutually exclusive (e.g. Ki67 and grade) and also face certain limitations Please consider using TNM staging instead. Lymph node metastases are not relevant as the vast majority of these patients are in stage 4 already, irrespective of the sensitivity of the imaging modality used to detect LN metastases. Please specify whether hepatic vs extrahep metastases grouping refers to GEP-NENS as lung NENs is another category and give lung mets first.
4. Please consider changing figures on OS and PFS analysis as previously highlighted.

Author Response

REVIEWER 2

The authors present a retrospective analysis of the use of CAPTEM regimen in a heterogeneous group of patients with NENs validating previous studies in this field. The data are of interest to the clinical community due to the largest so far cohort with real-world data, however there are some flaws that need to be addressed. The manuscript would benefit from further revision and clarification with respect to its limitations. In particular, subgroup analyses should be performed defining NEN population characteristics treated with CAPTEM more consistent with clinical practice.

Abstract
1. Define all abbreviations in the abstract: i.e.NEN, PFS, OS, ORR, pNEN, NEC, G1, G2.

All abbreviations have now been defined in the abstract
Comprehensive data are needed on the duration of CAPTEM study treatment in this study. There is currently a great controversy and a lack of evidence on fixed duration CAPTEM regimens vs. prolonged treatment until progression or toxicity occurrence. Please explain in the methods part of the abstract. This has been added to the methods section of the abstract. It now states that the average number of CATEM cycles was 9.5.
Safety analysis of CAPTEM should be added. This has been added to the abstract.
Please rephrase the conclusions and consider presentation of CAPTEM safety analysis. The sentence” The current study is the largest series to include 35 patients with non-pNENs” is not a study conclusion. It rather belongs to the discussion section. Safety analysis added as previously requested We have also deleted the following sentence: the current study is the largest series to include patients with non-pNENs

Introduction
This part is well-written and provides an overview in the field of available treatments including chemotherapy, and CAPTEM in particular, in NENs.

Methods
1. Page 3, line 108 and 110. Please provide references in right format.

This is not a reference, it is the web-based database that we used.
Please provide staging and grading system for different NEN primaries and relevant references in the methods section. Please explain how you managed patients grading before 2017, as the NEN G3 entity was introduces that year by the WHO classification for pNENs. Staging and grading system has been added, with relevant references All patients received a grade at time of study, post 2017. This has been explained in the paper now.
Did the study include central pathology review? No, although some (if not the majority) had review at our institution This is now addressed as a limitation in the limitation section
Please define baseline for survival estimates. We have defined baseline survival estimates and provided the appropriate reference. (Dasari, Arvind, et al. "Trends in the incidence, prevalence, and survival outcomes in patients with neuroendocrine tumors in the United States."JAMA oncology 10 (2017): 1335-1342)
As stable disease may be a favorable outcome particularly in NEN patients who received previous multiple lines of other treatments and progressed, please consider to provide data on disease control rate (DCR). We provided data on disease control rate This has been updated in the paper This has now been defined in the methods and reported throughout the results and referred to in the discussion section
Please use STROBE to report the findings of the analysis. Provide relevant analysis. STROBE checklist was reviewed and our methods section was re-written to include item numbers 4-9 and 11,12 Item 10 (“study size: Explain how the study size was arrived at”). This was not included in the methods section because it was previous described that all patients between November 2010 and June 2018.

Results
1. Please include descriptive data in the first two paragraph of the results’ section (page 4, lines 137-148) in Table 1 instead. The second paragraph is already a duplication.

The descriptive data in paragraph 1 and 2 have been summarized in table 1.
Please specify number of lines of treatment prior to CAPTEM administration and include this information in Table 1. This has been added to table 1
Page 5, section 3.2 on Treatment Response is very difficult to follow. Please use STROBE to report your results. This data should be mainly presented only in Table 2. WHO grade and Ki67 subgroup analysis are overlapping and one should be omitted as specified below (please see comments for tables). Results section re-written and reorganized Data has been primarily presented in table 2 WHO grade has been omitted
Please provide data on ORR and DCR also for the following strata: functionality (secretory status), FDG/Galium68-PET avidity, previous surgery, different treatments and specifically previous SSA, chemotherapy, PRRT and molecular targeted therapies. We did not collect information on gallium of FDG avidity, secretory status. However, we were able to provide ORR and DCR for patients who went through a prior surgery, SSA, chemotherapy, molecular targeted therapy, and prrt This has been added to the results section as a separate table (table 2b)

Please provide date on toxicities over time (when do they occur?) and overall discontinuation rates due to toxicities. Does previous chemotherapy increases the risk for serious toxicities with CAPTEM? We do not have data for when toxicities have occurred. As we are a tertiary center where patients are treated other places.  We have specific toxicities but we do not have dates of when this occurred. We have now added the rate of serious toxicities with previous chemotherapy and compared it to those patients who did not get previous chemotherapy. This has been added to the results section and has been added to the discussion portion of the paper as well.  
OS and PFS analysis of non-pNENs altogether is not comprehensive as this is a heterogenous group and colon NENs and UPOs are for example much more aggressive than SI-NENS and lung NENS. Please consider providing KM curves and log-rank analysis for each primary in two figures, one for PFS and one for OS. As this paper is an extension of the 2016 paper, we didn’t separate non-pnets.

Same applies to log-rank analysis and relevant figures for grade. Please use instead Ki67 index grouping and add the cut-off of 55% within G3 to address the difference in G3NEN vs. G3NEC (i.e 4 Ki67 groups) We removed WHO grade For figure 4, we used 4 groups for ki-67 as suggested
Multivariable Cox regression analysis should omit overlapping parameters and potentially include factors that I previously suggested to test in univariable analysis, i.e. functionality (secretory status), FDG/Galium68-PET avidity, previous surgery, different treatments and specifically previous SSA, chemotherapy, PRRT and molecular targeted therapies, number of previous lines of treatments. This is an important topic but better addressed in a prospective analysis

Discussion.
1. First paragraph in discussion should only present and elaborate on the study’s own results without making comparisons with other studies at this point. Safety analysis considerations and length of treatment aspects could be briefly discussed here.

Paragraph 1 now only discusses our own study findings
OS analysis conclusions are hard to make as these patients have received different line of treatments that preceded or followed CAPTEM. As such, the discussion in lines 283-284 is hard to comprehend. We have now addressed that patients received multiple treatment lines preceding captem therapy which would make comparison difficult
Line 301: The authors provide analysis results in 3 Ki67 categories and have put together G3NENs and NECs, so this claim is not true. Please consider providing complementary analysis to support these claims. Complementary analysis has been added to the results section to support these claims.
Safety analysis should be discussed with respect to studies on CAPTEM that the authors refer to. This has been added as a paragraph
Limitations: The study is limited by the potential selection bias toward long survivors and the heterogeneous population of patients without well-matched subgroups and control groups. Centralized radiological and histological assessment? These points have been added to the limitations section. It now reads Our study is also limited by the potential survivor treatment selection bias. Additionally, our population is heterogenous and would ideally have a matched control group.  Lastly, as our specialty center obtained patients from various medical practices, initial histological and radiological reports were generated by a variety of physicians outside of our treatment center.  As such, our study lacks centralized histological and radiological assessment.

Please specify future aspects that need to be addressed in conclusions. The conclusion addresses future research is needed specifically in NENs of varying ki-67 values and location (“Further research is needed to make definite recommendations regarding the use of this chemotherapy option in NENs of varying Ki-67 valuesand locations’)

Tables and Figures
1. Please explain why you present both grade and Ki67 proliferation index subgroup analysis in Table 2. Why is there a discrepancy in these groups? I would suggest to omit the grade grouping and use only the Ki67 one (add an extra subgroup there, i.e. ki67 20-55 and ki67>55). Lung NENs have other Ki67 cut-offs and other parameters such as type (typical/atypical) are more relevant in terms of prognosis. Using both grade and Ki67 grouping is actually confusing.

This has been previously addressed in your prior request Grade has been eliminated This has been addressed in our limitations. We recognized ki-67 is not a universally defined variable and lung papers use a variety of different ki-67 cut offs, but we wanted to add to the literature the use of captem in lung NENs.
Please apart from ORR, include DCR analysis in Table 2, as discussed above. DCR was added
Table 4: In univariable log-rank analysis the parameters selected are party mutually exclusive (e.g. Ki67 and grade) and also face certain limitations Please consider using TNM staging instead. Lymph node metastases are not relevant as the vast majority of these patients are in stage 4 already, irrespective of the sensitivity of the imaging modality used to detect LN metastases. We did not capture TNM We have eliminated the location of metastatic disease in the paper. The results were all similar except the data for liver metastasis; however, we had low number patients with liver metastasis so the results were likely not representative

Please specify whether hepatic vs extraheptic metastases grouping refers to GEP-NENS as lung NENs is another category and give lung mets first.

We felt it was best to eliminate the location of metastatic disease so I believe this request is not valid anymore
Please consider changing figures on OS and PFS analysis as previously highlighted. Based on our previous response to results request #6, we chose to not separate pNET vs non-pNET

Round 2

Reviewer 1 Report

No further comments, the revised paper is clearly improved over the origiginal Version!

Reviewer 2 Report

The authors have made changes to the manuscript as suggested by the Reviewers. I hereby confirm that the manuscript has been significantly improved following major revision and warrants publication in Cancers.